# * (Name Of This Paper Can Be Automatically Generated)

Matěj Lang*        Filip Kiraa Opálený        Palko Ulbrich        Lev Nikolajevič

Masaryk University

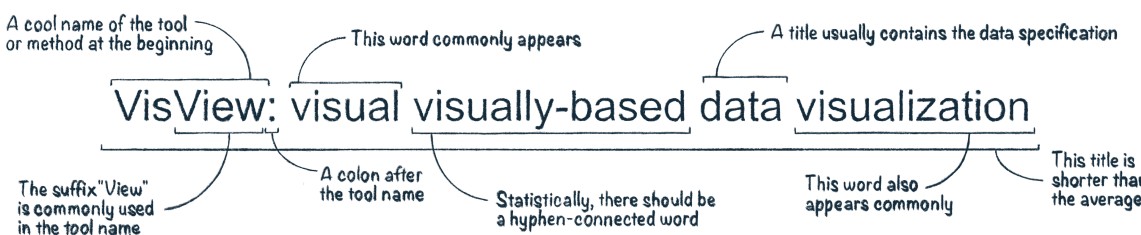

Figure 1: There are many recurring aspects in the names of IEEE VIS papers. We took them into consideration when designing a publicly available generator of random Vis titles.

## ABSTRACT

The scientific community produces many papers with dull and uninteresting titles. We wanted to explore how researchers break these standards and create memorable names. In this work, we present a coherent view of the current naming conventions of IEEE VIS papers and propose an online tool for their automatic generation. We took advantage of the predictable behavior of the naming habits of the vis community, and we compared them with cultural references and paper titles from other scientific fields. Based on our findings, we formed two iterations of vis naming grammar and used the second one in a publicly available tool. Lastly, inspired by similar works, we created a quiz comprising real and fake vis titles.

## 1 INTRODUCTION

One of the vital tasks in the research paper creation is the naming process. Unfortunately, in many cases, the title attempts are too generic and may discourage the reader from fully immersing themselves in the newly presented research. We wanted to aid with the process of inspiration for the visualization community, so we created the Vis paper title generator: a simple tool proposing the name of your next big publication.[1] The tool proposes a new research topic by following the paper naming conventions. The results were promising, but we needed further research to make them even more believable.

As the next step, we analyzed the original source of our inspiration, the actual paper titles. We took all titles of full papers presented at the IEEE VIS conference between 2011 and 2021. We performed a frequency analysis on the corpus and then searched for common patterns. We found several of them that are hinted at in the teaser (Fig. 1).

These findings were helpful for rebuilding our Vis name generator.[2] Among the improvements, we implemented a more elaborate way of creating tool names that often appear at the beginning of a title. We have also added a way of generating hyphenated words that show up in more than a third of the real titles. Finally, we have created a list of jargon words to improve the credibility of the titles.

*e-mail: langm@mail.muni.cz

All of these changes enhanced the final product, so we decided to put it to the test. We have created a small quiz for the reader to prove that our method can provide viable names.[3] We took a small set of real paper titles and mixed in a cherry-picked set of generated ones. It is at the discretion of the reader to compare the results and test their skill at distinguishing between the two.

With this work, we want to raise awareness of paper naming practices in the visualization community. We also want to provide a fun inspiration for creating memorable titles that will grab the reader's attention.

## 2 BACKGROUND

The process of creating random content to provide fun and inspiration is seen in many areas of culture. Tristan Tzara, one of the figures of the Dada movement, proposed cutting random words from a newspaper and arranging them to write poetry [29]. Other people have used variants of this exercise to create their own works of art (Fig. 2).

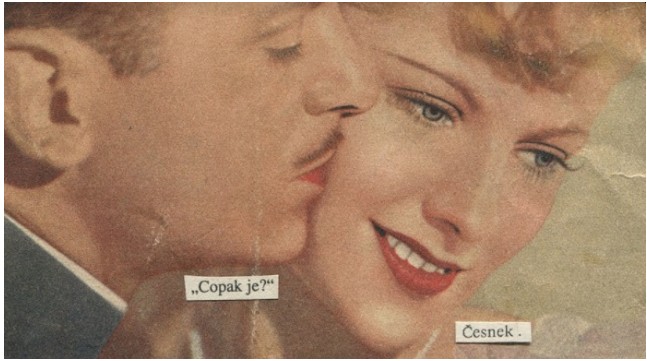

Figure 2: An example of a Czech phenomenon of "recycliterature," a way of creating funny or thoughtful collages out of discarded pictures and fragments of books [2].
Translation: −"What's wrong?" −"Garlic."

---

[1] https://xopaleny.pages.fi.muni.cz/chritmas-card/
[2] https://xopaleny.pages.fi.muni.cz/metapaper/

[3] https://xopaleny.pages.fi.muni.cz/vis-name-generator-quiz/

The (in)famous game Cards Against Humanity [1] asks players to finish a sentence with a card from their hand. The player does not necessarily choose at random, instead the most fitting phrase to make the funniest/weirdest/most unsettling sentence, but they choose from a very limited, random supply of choices.

We see other similar endeavors in projects such as name generators—a popular example of which is found at `https://www.fantasynamegenerators.com/`. It is a website providing randomly generated names for almost anything: names of various fantasy races, real names of people based on their country of origin, names for fictitious places—such as mage towers or gyms—and even names for your My Little Pony adventures. The list of all generators contains (to this date) impressive 1649 entries. It is often used by people writing stories, or players of role-playing games, such as Dungeons & Dragons, to find names for characters, places, or items.

With the rise of machine learning and especially neural networks, we can see a new trend in creating random content. The GANs (generative adversarial networks) provide a relatively easy way to extend the generation of random things into different media. Just provide a big enough dataset and train the network. The webpage This X Does Not Exist (`https://thisxdoesnotexist.com/`) provides a list of 40 pages that use a GAN to generate never before seen human faces, startup pages, pictures of horses, or voices. The neural networks behave in a black-box manner, only producing results without explaining the underlying principles. Therefore we opted not to use the machine learning approach, as we wanted to understand the naming conventions ourselves.

### 2.1 Naming Metastudies

Although the visualization field has not been scrutinized before regarding the titles, other fields have undergone this analysis. Philip Atkin has searched through 25 years (1976-2001) of papers from *British Medical Journal* for the phrases *paradigm shift* and *pushing the envelope* [5]. He found that the phrase *paradigm shift* has been on the rise since the mid-80s but has dropped significantly at the start of the millennium. The *envelopes* started to be *pushed* in the mid-90s and seemed to be on the rise. Atkin then warned that *pushing the envelope* might not be enough and that the authors should develop new exciting phrases and start *thinking outside the box*. The authors might have taken it too literally.

In the review of the title clichés by Neville Goodman of the same journal made ten years later, we can see the phrase *outside the box* has skyrocketed with 124 occurrences in just five years [15]. Goodman continues in Atkin's analysis: the *paradigms* are still happily *shifted* ten years later, while the *envelope-pushing* has faltered.

The authors find inspiration from various sources, clichés being only one of them. In another work of Neville Goodman, he explores the influence of pop culture on the paper titles [14]. Among the popular choices are the theatre plays of Shakespeare: *What's in a name?* (Romeo and Juliet), *To be or not to be* (Hamlet). That resulted in an unforgivable pun of ".*" [21] Other allusions include the works of Gertrude Stein *Rose is a rose is a rose*, Hans Christian Andersen with his *Emperor's new clothes*, or Brian Clark's *Whose life is it anyway?*

The movie titles and quotes also see frequent use. The blog post of Reto U. Schneider from 2010 [25] tracks several movie titles. The absolute winner is *The Good, the Bad, and the Ugly*, which showed up in at least 2700 paper names at the time. The variants include such gems as "The Good, the Bad, and the Cell Type-Specific Roles of Hypoxia Inducible Factor-1 in Neurons and Astrocytes" [31]. Then there are also variations on *Sex, Lies and Videotape* (526 variants), *Everything You Always Wanted to Know About Sex\* (\*But Were Afraid to Ask)* (513 variants), or *Some Like it Hot* (512 variants).

The music industry also has a considerable representation in the culture of paper naming. There is a long-running bet of a group of Swedish scientists to sneak in the lyrics of Bob Dylan [16]. The pa-

per by Gornitzki et al. explores the phenomenon of Bard's influence and expands its search to the whole field of biomedical literature. As a response, Brown et al. [10] published a letter with a list of references to Dylan's songs found in titles of meteorology papers. For a more detailed literary research, we kindly point the reader to the fantastic book *Academia Obscura* written by Glen Wright [32], that—among many other things—talks about bizarre and silly paper titles.

Not all scientific fields are equal in their ingenuity when it comes to paper titles. Mike Thelwall has extensively searched interesting titles across Scopus broad categories [27]. He has searched for common n-grams in 3.3 million articles, categorizing them into common sayings, song titles, books, etc. He found extreme discrepancies between scientific fields. On the one hand, Materials Science have roughly one in 29,000 articles that contain a common saying. On the other hand, Social Sciences have 1 in 900 names with common sayings on average. If we consider Visualizations as part of Computer Science, it sits somewhere in the middle with an average of 1 in 5,000. We believe that perhaps it is time to start *pushing the envelope* towards better results. The paper also lists the number of occurrences of each phrase. From that, we can see that the favorite stock phrase in the titles (and considered Optimal by some) is *more than meets the eye* with more than 250 occurrences. The movie titles are absolutely dominated by the previously mentioned "The Good, the Bad and the Ugly," with more than 350 uses. Contrast that with the second most used movie title, "How I Learned to Stop Worrying and Love" from *Dr. Strangelove*, with a measly 42 mentions.

### 2.2 Memorability & Attractiveness

We have discussed titles containing cultural references. Are they, however, making titles more memorable? What, exactly, makes the name stick? The work of Mahowald et al. [19] suggests that words with a clear one-to-one mapping between their form and meaning are the most memorable. E.g., the word *light* can mean "of low weight" or "a beam of photons." Furthermore, the *state of happiness* may be described as "happy," "cheerful," or "joyful." However, the word "pineapple" usually has one clear meaning mapped to a kind of fruit. That—in theory—makes it more memorable than the first examples.

Last year, Hallock et al. [18] studied the attractiveness of psychology paper titles. They have concluded that clichés do not significantly change the desire to read a title. However, this study did not include literary or other allusions, only standard figures of speech. The aspects that had the most significant positive impact included longer and more explanatory titles, colons splitting the title, posing a question that the paper answers, and spelling out unfamiliar acronyms.

### 3 PAPER TITLE GENERATOR—NAÏVE APPROACH

The original inspiration for the Vis paper name generator came from how GitHub generates random names for a new repository, a pair of *adjective-noun* selected from curated sets. This creates unique, easily memorable names that are sometimes unexpectedly funny.

The design of the first random generator is relatively simple. We have come up with a list of general words and sequences of words we are used to seeing on visualization papers, which we call intros. These include, but are not limited to: "visual representation," "comparative visualization," or "exploratory analysis." We also took a list of about 1000 English adjectives and ca. 6000 English common nouns (no given names, brand products, etc.).

The title is then built using one of the following templates:

```
${Adjective} ${intro} of/for/using ${noun}

${Intro} of/for/using ${adjective} ${noun}
```

We added two more templates to provide the user with more variability. The first is taking into account that we have seen many titles where the first word is the name of the tool, usually some kind

of a name containing "Vis" or "View." So we have created a list of words concatenated with a noun appearing later in the title.

```
${noun}${vis}: ${intro} of/for/using
${adjective} ${noun}
```

The second template creates the name of the tool as an acronym[4] Because not every acronym will be pronounceable, we perform a rudimentary check on the name. It is based on Markov chains, where every doublet and triplet of letters have assigned a score based on the probability of occurrence in a large corpus [33]. If the final score crosses the threshold, the acronym is used.

```
${ACRONYM}: ${Adjective} ${intro}
of/for/using ${noun}
```

Some examples of generated titles. We wrote a proposed research direction for illustration:

- **Tangible Dynamic Exploration for Goose.** This GCI (Goose-Computer Interaction) tool explores haptic feedback techniques for the aquatic fowl and its virtual space navigation.

- **Visual Guidance of Dramatic Spacing.** Sometimes the silence conveys the whole meaning. Our tool provides visual support to get the most out of those pregnant pauses.

- **ArchitectureViz: Temporal visualization of curious architecture.** The city holds thousands of stories within its walls that are not easily accessible to an average tourist. We created a time-lapse viewer of city development, from its inception as a small settlement to a sprawling metropolis.

- **VRONS: Visual representation of needless sailing.** Sailing is a very time-consuming activity that rarely serves any real purpose. This interactive map explores all the unnecessary sailboat rides and may serve to exploit eccentric billionaires.

## 4 DATA ANALYSIS

The next step in creating a better name generator is understanding what we are trying to emulate. For that, we have decided to perform the analysis of all IEEE VIS full papers presented at conferences between 2011 and 2021. For the analysis, we have used a Jupyter notebook [17] with the nltk (Natural Language Toolkit) package [7]. The whole notebook and the source data are available in our repository[5].

As a first task, we listed some title names to look for patterns. One of them became apparent quickly; the titles contain many hyphenated words. The corpus of 1755 individual titles contains 775 hyphens. The histogram (Fig. 3) shows their distribution in the titles. The maximum number of hyphenated words is six, and it belongs to this work of art: "KD-Box: Line-segment-based KD-tree for Interactive Exploration of Large-scale Time-Series Data" [35]. We further focused on the actual words that are connected by the hyphens. The frequency analysis reveals that the most used words are "multi-" (61 results) and "-based" (133 results) (Fig. 4).

The second interesting feature is the colons. There are 773 of them in the corpus, and from the histogram (Fig. 5), we can see that more than half of them are positioned right after the first word. A quick glance at the dataset shows that those first words are usually tool names. They can be generally split into two categories. The first is a word or a portmanteau of multiple words, often containing the word "Vis." This exquisite example speaks for itself: "PelVis: Atlas-based Surgical Planning for Oncological Pelvic Surgery" [26]. The second category is an acronym or initialism of the rest of the title, e.g., "YMCA - Your Mesh Comparison Application" [24].

---

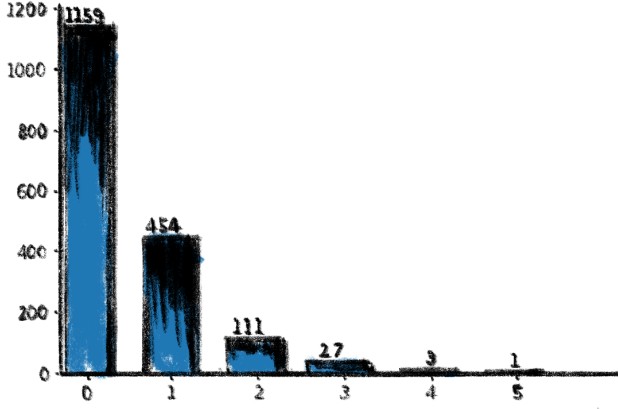

Figure 3: The histogram shows the number of hyphenated words in a title. There is a substantial amount of titles with at least one hyphenated word.

We took the subset of titles, where the first word is followed by a colon—presumably the name of the tool—and performed a substring analysis on that word. As expected, the most common substring is "vis," with 75 occurrences. Other popular words are also "lens," "graph," or "explore" (Fig. 6).

Next, to get the word frequency, we removed the stop words and performed the stemming operation to reduce the word to its root, e.g., "jumping"→"jump." We can see that the most frequent word, unsurprisingly, is "visual" with its derivatives "visualization," "visualizing," etc. Also, "data" and the root word "analy-" appear quite often.

The histogram of the title lengths based on the number of words gives us a nice approximation of a normal distribution, with an average title having around 9.75 words (Fig. 7). When we search for the most common word in each position, we can derive the truly average IEEE VIS paper title: "Visual Visual Visual Visual Visualization Visualization Data Visualization Data Dat." The last word is clipped on purpose.

In the last part of our analysis, we read through all 1755 titles to search for common phrases, clichés, and cultural references. In our search, we have found roughly 24 instances of phrases and figures of speech used in ordinary conversations. These include *looks good to me* "Looks Good To Me: Visualizations As Sanity Checks" [12], *at a glance* "At a Glance: Approximate Entropy as a Measure of Line Chart Visualization Complexity" [22], or *the curse of knowledge* "The Curse of Knowledge in Visual Data Communication" [34]. The phrase *state of the art*—considered a cliché by Goodman [15]—has come up three times; however, we argue that in the visualization field, it has become a technical term and part of jargon, so its inclusion is debatable.

When we compared our corpus to the work of Atkin [5], Goodman [15], and Thelwall [27], we found a surprising lack of clichés used in the titles. There was nothing *more than meets the eyes*, *one size* apparently *fits all*, and we have possibly *pushed the envelope* so far that we have hit *the end of the road*.

Similarly, as we have created a list of cultural references, we have found surprisingly little. There are 14 references in total, of which only one refers to the otherwise popular *The Good, The Bad, And The Ugly* "The Good, the Bad, and the Ugly: A Theoretical Framework for the Assessment of Continuous Colormaps" [11]. There is a mention of Matrix's *there is no spoon* "There Is No Spoon: Evaluating Performance, Space Use, and Presence with Expert Domain Users in Immersive Analytics" [6], Lord of the Rings' *One Ring to rule them all* poem "Obvious: A Meta-Toolkit to Encapsulate Information Visualization Toolkits — One Toolkit to Bind Them All" [13], and also Jurrasic Park "Jurassic Mark:

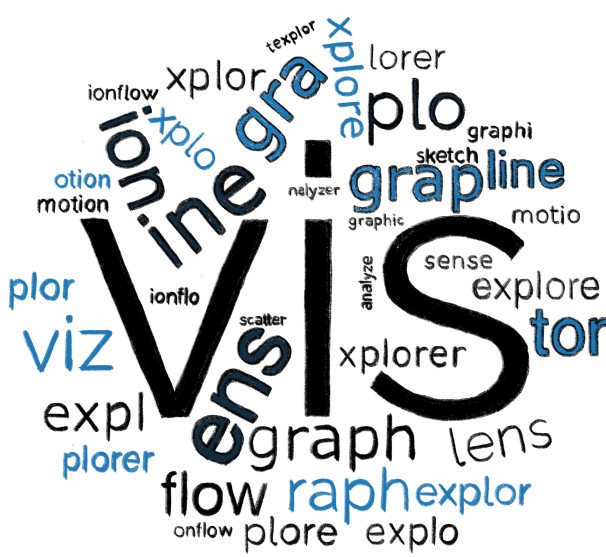

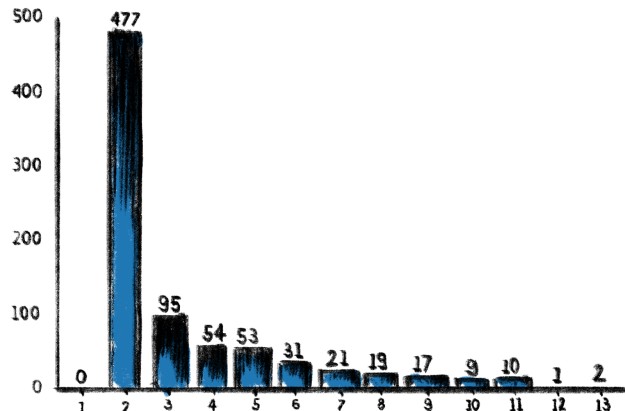

Figure 5: This histogram features the position of the first colon in a title. Each position is one word; that means no title starts with a colon. Many titles have a colon after the first word, indicating that the first word is the name of a tool or a method.

Figure 4: The wordcloud—being the superior visualization technique—shows the most frequent substrings in the names of the tools.

Inattentional Blindness for a Datasaurus Reveals that Visualizations are Explored, not Seen" [8].

The music scene is very poorly represented, with only one title. And although "YMCA - Your Mesh Comparison Application" [24] could also refer to Young Men's Christian Association directly, we have opted to include it in the music section instead, for the hit of 1978 by Village People.

The literary references are the only occurrence of two papers mentioning the same title: *What We Talk About When We Talk About Love*, a 1981 book by Raymond Carver: "What we talk about when we talk about data physicality" [20] and "What Do We Talk About When We Talk About Dashboards?" [23].

The other references include *Where's Waldo*, the series of British puzzle books by Martin Handford, "Finding Waldo: Learning about Users from their Interactions" [9], or "meet cute," the film and TV trope of the first meeting of future romantic partners "MeetCues: Supporting Online Meetings Experience" [4]. You can explore the complete, curated list of titles in our repository.

The tool names are the one place the authors are letting their imagination run wild. There are almost 500 titles with the first word as the name of the tool that sometimes gets very creative, e.g., "MeetCues: Supporting Online Meetings Experience" [4], "Stenomaps: Shorthand for Shapes" [30], or our favorite "PelVis: Atlas-based Surgical Planning for Oncological Pelvic Surgery" [26]. Tool names prove to be a positive deviation from otherwise somewhat dull titles. With these findings, we embarked on improving the random title generator model.

## 5 IMPROVED GENERATOR

The first step to making the title look more authentic is to use scientific jargon. We have achieved that using a curated list of nouns and adjectives that are common in the titles, e.g., "data," "machine learning," or "workflow" for the nouns, and "exploratory," "feature-based," or "interactive" for the adjectives. This allows for a wider variety in the names while keeping the number of base templates manageable.

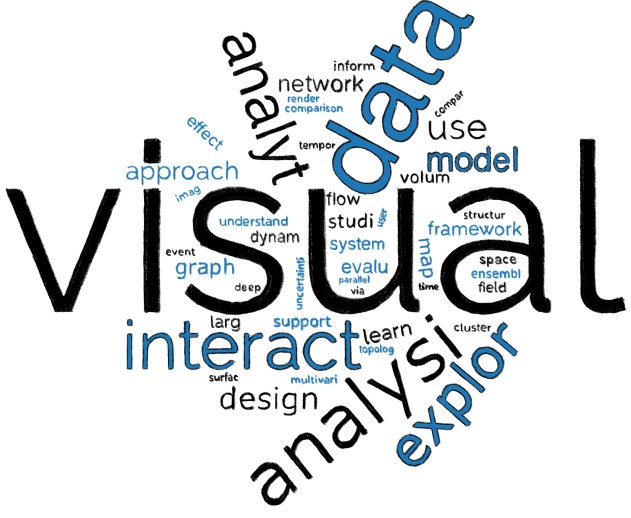

Figure 6: The most common word roots that appear in the paper titles. As expected, *visual* is on the top of the list.

Next is the use of hyphenated words. They appear commonly in the dataset, often with a predictable structure. We introduce them in two ways. First, we curated a set of jargon phrases. That way, we can somewhat control their credibility. The second way is to use the most used hyphenated words, such as "multi-" or "-based," and generate a word to complete it. This creates more varied results: "rails-based," "multi-snack," etc.

The third improvement came in the form of tool names. As previously stated, these are often subject to the most creative input in the whole title. We have devised a simple way to concatenate words to build interesting portmanteaus. After generating a noun, the list of all suffixes ("Vis," "Draw," "Paint," etc.) is searched. If there is a part of the noun that shares two letters with the start of the suffix, they get joined together. The results are uncanny: "Company"→"ComPaint," "Spectacle"→"SpectacLens."

For reference, we provide simplified, illustrative formal grammar for our new generations. Note that this grammar does not fully capture every intricacy, such as the details mentioned above of tool name generation.

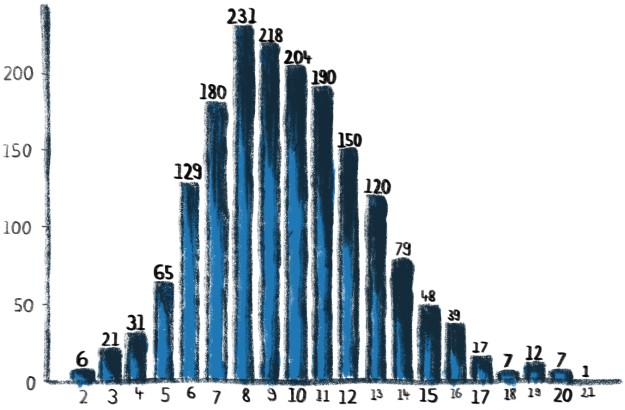

Figure 7: Histogram of title lengths. The most common is eight words per title, with an average of about 9.75 words per title.

$$
\begin{aligned}
\langle\text{paper-name}\rangle &\models \langle\text{towards}\rangle \mid \langle\text{adjVisword}\rangle \mid \langle\text{adjNoun}\rangle \\
&\quad \mid \langle\text{basedVisword}\rangle \mid \langle\text{multiVisword}\rangle \\
&\quad \mid \langle\text{star}\rangle \\
\langle\text{toolname}\rangle &\models \textit{pronouncable-acronym} \mid \textit{noun-toolpart} \\
&\quad \mid \textit{noun-toolpart-concatenated} \mid \lambda \\
\langle\text{filler}\rangle &\models \texttt{of} \mid \texttt{for} \mid \texttt{using} \\
\langle\text{any-noun}\rangle &\models \textit{noun} \mid \textit{jargon-noun}
\end{aligned}
$$

$$
\begin{aligned}
\langle\text{towards}\rangle &\models \langle\text{toolname}\rangle\ \texttt{Towards}\ \textit{jargonAdjective} \\
&\quad \textit{visword}\ \langle\text{filler}\rangle\ \textit{adjective noun} \\
\langle\text{adjVisword}\rangle &\models \langle\text{toolname}\rangle\ \textit{adjecitve visword}\ \langle\text{filler}\rangle\ \textit{noun} \\
\langle\text{adjNoun}\rangle &\models \langle\text{toolname}\rangle\ \textit{jargonAdjecitve visword}\ \langle\text{filler}\rangle \\
&\quad \textit{adjecitve noun} \\
\langle\text{basedVisword}\rangle &\models \langle\text{toolname}\rangle\ \langle\text{any-noun}\rangle\texttt{-based} \\
&\quad \textit{jargonAdjective visword}\ \langle\text{filler}\rangle\ \textit{adjective} \\
&\quad \textit{noun} \\
\langle\text{multiVisword}\rangle &\models \langle\text{toolname}\rangle\ \texttt{Multi-}\langle\text{any-noun}\rangle\ \textit{visword} \\
&\quad \langle\text{filler}\rangle\ \textit{adjective noun} \\
\langle\text{star}\rangle &\models \textit{adjective}\ \langle\text{any-noun}\rangle \\
&\quad \textit{visword}\texttt{:}\ \ \texttt{State of the Art}
\end{aligned}
$$

These three improvements create the backbone of our new, improved generator. Both the old version[6] and new[7] are available, so you can try to generate your own, compare them, and get inspired.

## 6 NAME QUIZ

As the final experiment, we created a quiz. We drew our inspirations from such masterpieces as *Who tweeted it: Donald Trump or Kanye West?* (Try to guess whether the tweet belongs to a former US president or a rap artist?), *IKEA or Death* (Is the following name a Finnish death metal group, or an article of the Swedish furniture manufacturer?), or *Antidepressants or Tolkien* (Does the name belong to a medication or a character from Tolkien's Middle Earth? This one is particularly tricky, as it contains characters from Silmarillion [28].).

---

[6] `https://xopaleny.pages.fi.muni.cz/chritmas-card/`
[7] `https://xopaleny.pages.fi.muni.cz/metapaper/`

The quiz has five cherry-picked real paper titles appearing at the IEEE VIS conference between 2011 and 2021 and five generated ones that were also chosen deliberately. You can try the quiz yourself; feel free to send us your results and suggestions.

## 7 CONCLUSION

We have created a viable method for creating new visualization paper titles. The original intent was to inspire the struggling researcher. We have also found that it provides a useful cognitive exercise for deciding which titles are utter nonsense and which might be worth exploring a little further. In any case, the tool might expand your horizons and inspire you to write the most memorable paper title.

Our literary research found that titles in the visualization community are often unremarkable. We believe it is time to get more creative. The tool names at the beginning of the titles are a positive deviation; we would like to encourage the researchers to keep up with this trend and explore it further.

### ACKNOWLEDGMENTS

The authors wish to thank Bára Kozlíková for her unbeknownst inspiration to create the generator and Sara Di Bartolomeo for her push to publish this work. We would also like to thank Glen Wright, whose book *Academia Obscura* served as an important foundation for the related work.

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
