# OpenReview forum: "* (This Name Can Be Automatically Generated)"
_IEEE.org/2022/Workshop/altVIS — Accept_

### Official Review · Reviewer_B7tx · 2022-08-08

**Review:**

In this paper the authors conduct a data analysis of paper titles from IEEEVIS in the last ten years which they use as a basis for a name generator. This name generator is available online in several different forms (including a quiz, which I regrettably only got 5/10 correct). I think this is a generally wonderful paper and I applaud the approach and execution. I especially liked the tone, the sketchy bar charts, and the quickly available interactive playground for the generator.

A handful of notes:

- I liked the data analysis quite well! I wished that there was a little bit more contextualization that might help reason about some of the findings, e.g. how many papers were published during that time? Are any of these findings trends? Why just the last 10 years and not a longer period, especially given the easy availability of this data (https://sites.google.com/site/vispubdata/home)? Did any of the trends follow the subconference splits (eg VAST/InfoVIS/SciVis)? Did the authors investigate any other sorts of bit
- The authors might find "Opening Sentences in Academic Writing: How Security Researchers Defeat the Blinking Cursor" to be relevant
- While the work is clearly already done, the authors might consider also investigating [tracery](https://tracery.io/) as a means to encapsulate their grammar.
- If im not mistaken the term cliche might be more clearly replaced with the slightly subtler term "snow clone", which captures both the memetic nature of these sorts of phrases but also indicates that they have a template nature.
- While I _strongly_ appreciate the intentional avoidance of machine learning in the design of the grammar, I think it might be interesting to feed some of the generated paper titles into a large language model, such as GPT3. For instance a moment of experimentation led me to the following

Title: Tangible Dynamic Exploration for Goose
Abstract: This paper presents a tangible user interface called Goose that supports dynamic exploration of digital data. Goose is based on a novel interaction technique called tangible data mapping, which allows users to directly manipulate digital data using physical objects. Goose was designed to support exploratory analysis of data, and to allow users to rapidly iterate through different data sets and visualizations. To evaluate Goose, we conducted a user study with eight participants. The results showed that Goose was effective for exploratory data analysis, and that participants were able to quickly generate different visualizations and data sets.

- "prof or Hobo" and the language around it is not inclusive, and should be replaced with an example that doesn't draw humor from some societies most vulnerable members. (or just be removed, the other examples are sufficient!)
- A minor typo: I think all of the forward quotes are backwards? E.g. "example quote" should be replaced with ``example quote'' in latex


**Conflicts:**

I do not believe I have any conflicts.

**Review Inclusion:**

Yes

**Sufficiently Alt:**

Yes

**Superlative:**

Most Generative

---

### Official Review · Reviewer_uGnC · 2022-08-23

**Review:**

This rules. Very fun, very alt, raises a good point (about how many vis papers [titles] are formulaic and uninteresting, except for the few cases where people get a chance to go nuts coining system names).

Not much to say beyond that. Accept it.

Some issues to fix:

1) excentric -> eccentric
2) Double quotes should be `` in latex, rather than ‘’ (which generates closing quotes)

Other thoughts:

1) While it's true that [9] is a reference to Where's Waldo, the task is also literally to find Waldo using examples from various Waldo books, so…

2) Not sure how to feel about having two of my own papers in the "cliche/reference" categories get mentioned by name in the paper body. I guess I need to be either more creative or much, much, less creative in naming papers.

3) I Wish the quiz wasn’t quite so cherry-picked, would be interesting to see what the baseline outputs of the models were, rather than just the creme of the crop.

4) The quiz is almost exactly a GAN setting, so would be interesting to see if the formal grammar, or a less constrained GAN model would generate higher quality paper titles (in terms of being able to distinguish them from the genuine article).

5) Another thing I am reminded of is the (tongue in cheek) paper "Paper Gestalt" by the pseudonymous Carven von Bearnensquash, where they try to predict if a paper will be accepted based on appearance alone, and find that features like colorful figures, using up the full page space, and having complex formula all seem to produce better outcomes. The (ever so slightly) less tongue-in-cheek followup work "Deep Paper Gestalt" by Jia-Bin Huang extends this model using ML, and finds it to be pretty selective (although it, somewhat ironically, results in a model that predicts rejection for the paper talking about the model). This is the clear next step for this work, to be able to connect paper titles and visual elements in visualization papers to some predictive outcome. I hope we can get our own automatic Sokol hoaxes within the next few years.

**Conflicts:**

No conflicts as far as I know.

**Review Inclusion:**

Yes

**Sufficiently Alt:**

Yes

**Superlative:**

Most Human-Centric Insight-Based Crowd-Sourced Evaluative Exploration of the State of the Art

---

### Official Review · Reviewer_o3Gr · 2022-08-24

**Review:**

The authors endeavor to provide memorable titles by innovatively providing visualizations for the titles.
Pros:
1. the author's angle to visualize the title of a paper in action is innovative
2. the research provides a comprehensive literature review for automatically generated contents in other occasions
3. the study proposes a viable method for creating visualization paper titles.

Cons:
the quality or at least resolutions of the illustrations could be improved



**Conflicts:**

NA

**Review Inclusion:**

Yes

**Sufficiently Alt:**

Yes

**Superlative:**

Most practical

---

### Official Review · Reviewer_JeyR · 2022-08-30

**Review:**

Meta Review:

The paper is humorous, while insightful and thoughtfully considered. References are abundant, relevant, and engaging. Please pay special attention to formatting suggestions and recommendations about inclusive language from other reviewers. I would recommend a strong accept for this delightful paper.

See also: for section 6, an additional reference - https://www.cnn.com/interactive/2016/02/politics/trump-kanye-who-tweeted/

**Conflicts:**

A colleague from my lab is listed in the acknowledgements, but I was not aware of this connection until completing my own reading of the paper and reading the other reviews.

**Review Inclusion:**

Yes

**Sufficiently Alt:**

Yes

**Superlative:**

Most formulaic title

---

### Decision · Program_Chairs · 2022-08-31

Accept